# Synthesis and Spectroscopic Characterization of Selected Water-Soluble Ligands Based on 1,10-Phenanthroline Core

**DOI:** 10.3390/molecules29061341

**Published:** 2024-03-18

**Authors:** Jacek E. Nycz, Natalia Martsinovich, Jakub Wantulok, Tieqiao Chen, Maria Książek, Joachim Kusz

**Affiliations:** 1Institute of Chemistry, Faculty of Science and Technology, University of Silesia in Katowice, ul. Szkolna 9, 40-007 Katowice, Poland; 2Department of Chemistry, University of Sheffield, Sheffield S3 7HF, UK; n.martsinovich@sheffield.ac.uk; 3Ministry of Education Key Laboratory of Advanced Materials for Tropical Island Resources, Hainan Provincial Key Laboratory of Fine Chem, Hainan Provincial Fine Chemical Engineering Research Center, Hainan University, Haikou 570228, China; chentieqiao@hnu.edu.cn; 4Institute of Physics, Faculty of Science and Technology, University of Silesia in Katowice, 75 Pułku Piechoty 1a, 41-500 Chorzów, Poland; maria.ksiazek@us.edu.pl (M.K.); joachim.kusz@us.edu.pl (J.K.)

**Keywords:** phenanthroline, water-soluble ligand, dithiocarboxyl, hydroxydialdehyde, hydroxydicarboxylic acid, 10-hydroxybenzo[*h*]quinoline-7,9-dicarboxylic acid, 9-methyl-1,10-phenanthroline-2-carboxylic acid, vinyl, styryl

## Abstract

Water-soluble ligands based on a 1,10-phenanthroline core are relatively poorly studied compounds. Developing efficient and convenient syntheses of them would result in new interesting applications because of the importance of 1,10-phenanthrolines. In this manuscript, we describe novel and practical ways to introduce a carboxyl and, for the first time, a phenol and dithiocarboxyl group under mild reaction conditions. This strategy enables highly efficient and practical synthesis of suitable organosulfur compounds with high added value, high chemoselectivity, and a broad substrate range. We present the selective conversion of a hydroxydialdehyde in the form of 10-hydroxybenzo[*h*]quinoline-7,9-dicarbaldehyde into its derivative, unique hydroxydicarboxylic acid, by an oxidation procedure, giving 10-hydroxybenzo[*h*]quinoline-7,9-dicarboxylic acid. A similar procedure resulted in the formation of 9-methyl-1,10-phenanthroline-2-carboxylic acid by oxidation of commercially available neocuproine. An alternative method of obtaining 1,10-phenanthroline derivatives possessing carboxylic acid group can be based on the hydrolysis of ester or nitrile groups; however, this synthesis leads to unexpected products. Moreover, we apply Perkin condensation to synthesize a vinyl (or styryl) analog of 1,10-phenanthroline derivatives with phenol function. This reaction also demonstrates a new, simple, and efficient strategy for converting methyl derivatives of 1,10-phenanthroline. We anticipate that the new way of converting methyl will find wide application in chemical synthesis.

## 1. Introduction

Water-soluble ligands are widely used in various fields, including chemistry, biochemistry, and coordination chemistry. They play a crucial role in many chemical and biological processes and industrial applications, such as hydrogenation and hydrogen production from water [1], oxidation reactions [2], and hydroformylation [3,4]. The chemical and physical properties of water-soluble 1,10-phenanthroline derivatives with sulfonate, carboxylic, phosphonic acid, or hydroxylic groups in their constitution have been widely studied and used for the separation of metallic atoms and capillary electrophoresis, as well as for the development of bio-inorganic probes [5]. Furthermore, these hydrophilic groups’ presence allows researchers to conduct in vitro cytotoxicity tests and assess the compounds’ biological activity [6]. In addition, complexes with 1,10-phenanthroline derivatives functionalized with sulfonate, carboxylic, phosphonic acid, or hydroxylic groups are capable of binding to metal oxide surfaces [7]. Studies have already indicated the broad scope of catalysts based on water-soluble ligands, demonstrating the industrial importance of catalyzed reactions in aqueous systems [8]. Historically, reducing molecules containing unsaturated carbon–carbon or carbon–oxygen bonds was the first application of a water-soluble catalyst [8,9,10,11,12]. In addition, water-soluble metal complexes have gained interesting applications in biochemistry. Some water-soluble phenanthroline complexes of rhodium, iridium, and ruthenium have found applications for the regeneration of NADH [13].

1,10-Phenanthroline is one of the most explored chelate systems among *N*-heterocyclic compounds due to its robust redox stability and a wide range of complexing properties [14,15]. These properties result from the presence of two nitrogen atoms always in a characteristic alignment in the rigid molecule structure imposed by the central benzene ring, which allows it to act as a bidentate ligand, which wraps around the metal ion, providing increased stability. 1,10-Phenanthroline can form stable complexes with various metal ions in different configurations at various oxidation states depending on conditions [16,17]. These complexes have characteristic electronic [18] and antimicrobial [19] properties, which are being continuously exploited in various fields of physics, chemistry, biology, biochemistry, crystallography, and medicine.

Water-soluble derivatives of 1,10-phenanthroline can provide many valuable applications and are important reagents for green chemistry, which is why research is essential to obtain new applications for them and new synthetic methods. So far, the water-soluble 1,10-phenanthrolines described in the chemical literature are divided into three classes: carboxylic, sulfonic, and phosphonic acid derivatives (Table 1). The described acidic 1,10-phenanthroline derivatives are bifunctional compounds that imitate amino acids or naturally occurring nicotinic acid. Unfortunately, the methods reported in the literature have many limitations, and the presented molecules are poorly characterized, especially for phosphonate and sulfonate derivatives. Sulfonate derivatives such as 4,7-diphenyl-1,10-phenanthroline disulfonic acid disodium salt or 2,9-dimethyl-4,7-diphenyl-1,10-phenanthroline disulfonic acid disodium salt are commercially available, but only as a mixture of regioisomers. There are no described synthesis procedures for the mentioned class of compounds in the literature. To the best of our knowledge, only one paper reported their NMR characteristics (only as a mixture of regioisomers) [6]. Their synthesis procedures are only presented in patent applications [20,21,22]. Beletskaya et al. published a synthesis method of (1,10-phenanthrolin-3-yl)phosphonic acid as an example of only one phosphonate representative of acidic 1,10-phenanthroline derivative (Table 1) [23].

The current study presents the synthesis and spectroscopic characterization of selected water-soluble ligands based on a 1,10-phenanthroline core, rationalized based on density functional theory (DFT) calculations.

## 2. Results and Discussion

Existing publications and patents presenting the syntheses of derivatives of 1,10-phenanthroline carboxylic acid can be divided into three main methods. These are shown in Figure 1.

Most of the previously reported syntheses of carboxylic acid derivatives of 1,10-phenanthroline are based on multi-step chemical transformations, which have the disadvantages of harsh reaction conditions and toxic materials, relatively expensive reagents or catalysts, poor product yields, long reaction times, and tedious separation procedures (Figure 1) [7,24,27]. The inaccessibility of convenient synthetic methodologies for water-soluble 1,10-phenanthrolines is probably the most crucial reason for the lack of their applications. To meet the demand, we developed synthesis methods of selected water-soluble 1,10-phenanthrolines without employing highly toxic salts of transition metals. These methods avoid the additional handicaps of transition metals, such as the possibility of destructive oxidation of the initial 1,10-phenanthroline skeleton and contamination of the product compounds or their derivatives with transition metals, which may affect the results obtained in biological studies.

### 2.1. Synthesis of Carboxylic Acid Products by Oxidation of Methyl Groups

We develop an alternative procedure to method A presented in Figure 1 that is based on the oxidation of commercially available 2,9-dimethyl-1,10-phenanthroline (neocuproine; **1a**) by means of sodium chlorite (Figure 2). The reaction occurs in mild conditions in water with the use of an excess of sodium chlorite. The reaction is carried out initially at room temperature, and then the suspension is heated to reflux temperature with constant stirring for ten hours. The crucial step is the isolation of the crude product. For this purpose, the reaction mixture was carefully acidified with 10% HCl solution until the pH was acidic (pH = 2.8). This procedure was chosen based on the low cost of its implementation and the availability of reagents. As a result of the reaction, 9-methyl-1,10-phenanthroline-2-carboxylic acid (**2a**) was selectively synthesized in the case of a minor reaction scale, up to 1.0 g, with 58% yield. However, on a higher reaction scale, the reaction is more complicated (Figure 2). We isolated and identified chlorinated products **2b** (with 18% yield), **2c**, and **2d**, similar to Miron and co-workers’ results [26]. Products **2c** and **2d** were only identified by MS techniques. During the acquisition of NMR spectra in 2M water-*d*_2_ KOD solution/DMSO-*d*_6_, all products showed pH-dependent regioselective isotopic H/D exchange reactions (Figure 2). The identification of chlorinated products **2b**, **2c**, and **2d** instead of compound **2a** suggests that in the first step, the 2-methyl-9-(trichloromethyl)-1,10-phenanthroline or a similar intermediate is formed, similar to Miron et al. [26], which is further chlorinated, while in the next step, during the hydrolysis reaction, the final products are formed. During the reaction of sodium chlorite (NaClO_2_), hypochlorous acid (HOCl) can be released, analogous to the Pinnick reaction. The HOCl byproduct, a reactive oxidizing agent, can also cause other side reactions with the organic materials. For example, HOCl can react with double bonds in the organic starting reactant or product via a halohydrin formation reaction. The presence of molecule **2b**, **2c**, and **2d** with a chlorine atom at the C4 (or C7) position suggests a mechanism of the S_N_Ar type, for which no example exists in the literature. The carboxyl and methyl groups’ simultaneous presence in the structure of the described acids can potentially enable further chemical transformations and allow for further functionalization.

### 2.2. Oxidation of Aldehydes

We checked the effectiveness of our procedure for oxidation of selected hydroxy dialdehydes, i.e., 8-hydroxy-2-methylquinoline-5,7-dicarbaldehyde (**1b**) and 10-hydroxybenzo[*h*]quinoline-7,9-dicarbaldehyde (**3**) (Figure 1 and Figure 3). Hydroxy dicarboxylic acids are an intriguing research topic, as they can be expected to participate in a variety of reactions due to the simultaneous presence of carboxyl and hydroxyl groups. Due to the site for the donor force and the excellent chelating properties associated with the presence of a hydroxyl group in the vicinity of the nitrogen atom, the compounds described can gain applications as ligands in coordination chemistry and complexometric chemical processing. The 8-hydroxy-2-methylquinoline-5,7-dicarbaldehyde (**1b**) (Figure 3) and 10-hydroxy-benzo[*h*]quinoline-7,9-dicarbaldehyde (**3**) (Figure 1) were oxidized in similar reaction conditions to those in the above experiment presented in Figure 2. The oxidation of aldehyde **3** leads to 10-hydroxybenzo[*h*]quinoline-7,9-dicarboxylic acid (**3a**) with 32% yield (Figure 1). The aldehydes with different structures were chosen to test the possible limitations of the reaction proposed in this study. The reaction was initially carried out under mild conditions at room temperature until gas evolution ceased and then at a temperature of 100 °C. To achieve better solubility, the solvent used in the reaction was water for the quinoline derivative and formic acid for molecule **3**.

The reaction of dicarbaldehyde **1b** with sodium chlorite did not lead to an expected product (Figure 3). The reaction resulted in breaking one of the C-C bonds in the phenol ring and further side reactions that we recently described [32]. Density functional theory calculations of reactant **3** in Figure 1 and reactant **1b** in Figure 3 (Appendix A) showed a longer C6-C7 bond in molecule **1b** (1.410 Å) than in compound **3** (1.391 Å), potentially making the C6-C7 bond in **1b** easier to break. Calculations of the expected product (diacid) in Figure 3 similarly showed a longer C6-C7 bond (1.408 Å) and C7-C8 bond (1.402 Å) compared to the product **3a** in Figure 1 (1.395 and 1.396 Å, respectively), showing that these bonds in the expected product in Figure 3 would be relatively easy to break. The obtained products were characterized by spectroscopy and mass spectrometry. These results are also consistent with the work of Miron et al. [26]. The chlorination in the C6 position in the quinoline constitution, followed by hydrolysis, can explain the ring-opening reaction.

As shown above, the use of H_2_O_2_ (30%) in methanol or Tollens’ reagent did not allow us to obtain the target dicarboxylic acid, or the product was unstable. Therefore, we decided to use HNO_3_, as in earlier work by Cian et al. Hureau et al. [25,31]. Again, we did not obtain the expected product, but we obtained 2-methyl-5,7-dinitroquinolin-8-ol in 65% yield. This suggests that either each aldehyde group was oxidized by nitric acid to the carboxylic acid, which further underwent decarboxylation, and the resulting in situ 2-methylquinolin-8-ol was then nitrated, or, more likely, each aldehyde group was oxidized to nitrate ester, which underwent rearrangement (Figure 3). It is worth mentioning that the resulting 2-methyl-5,7-dinitroquinolin-8-ol and 8-hydroxy-2-methylquinolin-5,7-dicarboxylic acid have similar *m*/*z* values of 249 and 247, respectively.

### 2.3. Hydrolysis of Ester- and Nitrile-Substituted Phenanthroline Compounds

Another possibility for introducing the carboxylic acid group into a 1,10-phenanthroline core can be the hydrolysis of precursors containing an ester or nitrile group. This creates a convenient opportunity for the presence of other groups suitable for further transformations. For this purpose, we hydrolyzed selected derivatives of 1,10-phenanthroline that contain either chlorine atoms or 9*H*-carbazol-9-yl and 10*H*-phenoxazin-10-yl substituents at C4 and C7 and methyl or ethyl esters or nitrile group at C5, which can be further hydrolyzed.

The 4,7-dichloro-1,10-phenanthrolines with a carboxylic acid group as substituent should be readily soluble in water. We already reported that the hydrolysis of the nitryl group in 4,7-dichloro-1,10-phenanthroline-5-carbonitrile (**1e**) using an excess of hydrochloric acid leads to the formation of not the expected 4,7-dichloro-1,10-phenanthroline-5-carboxylic acid but the unexpected 7-chloropyrrolo[2,3,4-*de*][1,10]phenanthrolin-5(4*H*)-one, which was isolated with 37% yield (Figure 4) [32]. This may be attributed to the formation of a 4,7-dichloro-1,10-phenanthroline-5-amide intermediate, where the amide group, instead of being further hydrolyzed to carboxylic acid, acts as a nucleophile and attacks the nearby C4, with the chloride at C4 acting as a leaving group.

Further unexpected results were obtained during the reaction of 4,7-disubstituted-1,10-phenanthroline-5-carbonitriles (**1f,g**) in an alkaline water environment (Figure 4). The 4,7-di(9*H*-carbazol-9-yl)-1,10-phenanthro line-5-carbonitrile (**1f**) and 4,7-di(10*H*-phenothiazine-10-yl)-1,10-phenanthroline-5-carbonitrile (**1g**) transform selectively to 4,7-di(9*H*-carbazol-9-yl)-9-oxo-9,10-dihydro-1,10-phenanthroline-5-carbonitrile (**4a**) and 9-oxo-4,7-di(10*H*-phenothiazin-10-yl)-9,10-dihydro-1,10-phenanthroline-5-carbonitrile (**4b**) (Figure 4). We hypothesize that these products were formed through a reaction of oxidative nucleophilic substitution of hydrogen (ONSH) of the type S_N_ArH, with the hydroxide ion as a nucleophile in the presence of air as an oxidized reagent. In both cases, hydroxide ions attack only at C9 rather than at the C2 position. The preference for the C9 site may be explained by the difference in charges on atoms C2 and C9, caused by the non-symmetric position of the CN group: according to the DFT calculations, Mulliken charges on C9 are slightly more positive than those on C2 (by 0.01–0.004 e) both in molecules **1f** and in **1g**, making the C9 position slightly more susceptible to nucleophilic attack.

The hydrolysis reaction of 4,7-dichloro-2,9-dimethyl-1,10-phenanthroline-5-carbonitrile (**1c**) or methyl 4,7-dichloro-2,9-dimethyl-1,10-phenanthroline-5-carboxylate (**1d**) in an alkaline environment leads to their precursor 4,7-dioxo-1,4,7,10-tetrahydro-1,10-phenanthroline-5-carbonitrile or methyl 2,9-dimethyl-4,7-dioxo-1,4,7,10-tetrahydro-1,10-phenanthroline-5-carboxylate, respectively (Figure 4). The chlorine atoms in the C4 and C7 positions are activated and easily substituted with hydroxyl in the alkaline environment. The work of J. King-Underwood et al. similarly presents reactions in the alkaline environment where the hydrolysis of the chlorine atom occurs [33].

Methyl 4,7-dichloro-2,9-dimethyl-1,10-phenanthroline-5-carboxylate (**1d**) crystallizes in the monoclinic system, *P*21/c, with unit cell dimensions of a = 21.0882(6) Å, b = 11.6270(3) Å, c = 7.0683(2) Å, V = 1728.66(8) Å^3^, and Z = 4; the monoclinic angle (β) is 94.101(3)°, and the unit cell contains four molecules of ester and twelve of water (Figure 1c). The distance between the centroids of the phenyl and pyridine rings is in the range of 3.755 to 4.342 Å (Figure 1b).

### 2.4. Synthesis of Dithiocarboxyl Derivatives

The introduction of a dithiocarboxyl group into a 1,10-phenanthroline core is unknown in the chemical literature. We report it here for the first time (Figure 5). We presented only products with newly formed C4-CSSH bonds. The attempts to isolate other products with CH_2_-CSSH bonds in pure form for analytical measurements were unsuccessful. The largest difference in C-13 NMR spectroscopy is visible by comparing the carboxylic group of molecules **2a**, **2b**, and **3a** and the dithiocarboxylic group for dithiocarboxylic acid **5a**. Similarly to our previous paper [34,35], we observed an “unusual” chemical shift of 248 ppm for the dithiocarboxylic acid group of 2,9-dimethyl-1,10-phenanthroline-4-carbodithioic acid (**5a**). Density functional calculations of C-13 NMR spectra of 2,9-dimethyl-1,10-phenanthroline-4-carbodithioic acid showed the chemical shift of 243 ppm vs. TMS for the carbodithiolic carbon, in very good agreement with the experimental peak of 248 ppm. An even larger shift of 274 ppm was calculated for the carbodithiolate carbon in the zwitter-ion form of product **5a**. This value is much larger than the 171 ppm chemical shift calculated for the carboxylic carbon in the carboxylic acid analog. The relative deshielding of anions is observed in other cases as well. It is likely to result from an increased paramagnetic shielding contribution (the paramagnetic shielding is negative, so the nucleus is deshielded when this contribution grows), which itself is related to a decrease in the HOMO-LUMO gap. This is confirmed by our DFT calculations of molecular orbitals (Appendix A), which show a decrease in the HOMO-LUMO gap of the dithioacid (3.13 eV) compared to acid (4.31 eV, and an even smaller HOMO-LUMO gap for the zwitter-ion form of the dithioacid (1.25 eV). This change in the HOMO-LUMO gap can be attributed to changes in the localization of the frontier orbitals: in the carboxylic acid, both HOMO and LUMO are localized on the phenanthroline group, while in the dithioacid, the HOMO is localized on the thioacid group and the HOMO-1 on the phenanthroline group; in the dithioacid zwitter-ion, HOMO to HOMO-2 are on thiolate group, and HOMO-3 is on the phenanthroline group. As a result, the increased paramagnetic shielding acts as an additional magnet (Figure 2).

### 2.5. Synthesis of Vinyl Derivatives

The synthesis of vinyl analogs of 1,10-phenanthroline derivatives could be an alternative solution to introducing hydrophilic functions such as phenol or carboxylic acid groups. In the literature, only a few papers deal with the synthesis of vinyl analogs of 1,10-phenanthroline [28,36]. Kathirgamanathan et al. efficiently obtained four analogs as electron-transporting materials in organic light-emitting diodes (OLEDs). In a series of experiments, we apply Perkin condensation to get four vinyl analogs of 1,10-phenanthroline, which also possess phenol groups (Figure 6). The synthetic procedure was based on our early work [37,38].

The 4,7-dichloro-1,10-phenanthrolines are sensitive to hydrolysis, which we mentioned above. This can explain why, during the synthesis of vinyl analogs of molecule **1h**, we obtain 5-methyl-1,10-dihydro-1,10-phenanthroline-4,7-dione derivatives **6a** and **6b**, which possess the carbonyl group instead of chloride atom. Wanting to avoid a hydrolysis reaction of chloride atoms located in the C4 and C7 positions, in the next experiment, we chose neocuproine **1a** as a simpler model (Figure 7). We efficiently obtained styryl **6d** and its precursor **6c** by employing a straightforward and effective procedure.

## 3. Materials and Methods

### 3.1. Materials

All experiments were carried out in an atmosphere of dry argon, and flasks were flame dried. Solvents were dried by usual methods (diphenyl ether, diethyl ether, and THF over benzophenone ketyl, CHCl_3_, and CH_2_Cl_2_ over P_4_O_10_, hexane, and pyridine over sodium–potassium alloy) and distilled. Chromatographic purification was carried out on silica gel 60 (0.15–0.3 mm, Macherey-Nagel GmbH & Co. KG, Dueren, Germany). Carbon disulfide (CS_2_), 2,9-dimethyl-1,10-phenanthroline (neocuproine), 2,4-dihydroxybenzaldehyde, 2-hydroxybenzaldehyde, silver nitrate (AgNO_3_), *t*BuLi (1.7 M in pentane), and sodium chlorite (NaClO_2_) were purchased from Sigma–Aldrich (Poznań, Poland) and were used without further purification. Synthesis of 8-hydroxy-2-methylquinoline-5,7-dicarbaldehyde and 10-hydroxybenzo[*h*]quinoline-7,9-dicarbaldehyde followed our procedure described in the literature [39].

### 3.2. Instrumentation

NMR spectra were obtained with Avance 400 and 500 spectrometers (Bruker, Billerica, MA, USA) operating at 500.2 or 400.2 MHz (^1^H) and 125.8 or 100.6 MHz (^13^C) and 202.47 (^31^P) at 21 °C. Chemical shifts were referenced to ext. TMS (tetramethylsilane) (^1^H, ^13^C) or 85% H_3_PO_4_ (^31^P), or using the residual CHCl_3_ signal (δ_H_ 7.26 ppm) and CDCl_3_ (δ_C_ 77.1 ppm) as internal references, and ext. DSS for ^1^H and ^13^C-NMR, respectively. Coupling constants are given in Hz. The LCMS-IT-TOF analysis was performed on an Agilent 1200 Series binary LC system coupled to a micrOTOF-Q system mass spectrometer (BrukerDaltonics, Bremen, Germany). High-resolution mass spectrometry (HRMS) measurements were performed using a Synapt G2-Si mass spectrometer (Waters, New Castle, DE, USA) equipped with an ESI source and quadrupole-time-of-flight mass analyzer. To ensure accurate mass measurements, data were collected in centroid mode, and mass was corrected during acquisition using leucine enkephalin solution as an external reference (Lock-Spray) (Waters, New Castle, DE, USA). The measurement results were processed using the MassLynx 4.1 software (Waters, Milford, MA, USA) incorporated within the instrument. A Nicolet iS50 FTIR spectrometer was used to record spectra in the IR range of 4000–400 cm^−1^. FTIR spectra were recorded on a Perkin Elmer (Schwerzenbach, Switzerland) spectrophotometer in the spectral range of 4000–450 cm^−1^ with the samples in the form of KBr pellets. Elementary analysis was performed using a Vario EL III apparatus (Elementar, Langenselbold, Germany). Differential scanning calorimetry (DSC) measurements were performed using a Q2000 calorimeter (TA Instruments, New Castle, DE, USA) in a nitrogen stream at a scanning rate of 10 °C/min. Samples were analyzed in aluminum pans in the temperature range of 50 to 350 °C. Melting points were determined on an MPA100 OptiMelt melting point apparatus (Stanford Research Systems, Sunnyvale, CA, USA) and were uncorrected. UV/Vis absorption spectra were measured on a SHIMADZU (Tokyo, Japan) UV-VIS-NIR spectrophotometer using quartz (Suprasil) cuvettes (10 mm path length). Fluorescence spectra were recorded on an FLS-900 spectrofluorimeter (Edinburgh Instruments, Edinburgh, UK) and were measured in quartz (Suprasil) cuvettes (10 mm path length) at 25 °C. The phosphorescence spectrum was recorded on an FLS-900 spectrofluorimeter (Edinburgh Instruments, Edinburgh, UK) at 77 K using a 3 mm (inner diameter) quartz tube inside a quartz liquid nitrogen Dewar flask.

### 3.3. X-ray Diffraction Experiments

The data for **1d** were collected using a SuperNova diffractometer (Agilent Technologies, Santa Clara, CA, USA, currently Rigaku Oxford Diffraction). Accurate cell parameters were determined and refined using the CrysAlis^Pro^ program [40]. Also, the integration of the collected data was performed with this program. The structure was solved using direct methods with the SHELXS-2013 program and then refined using the SHELXL-2019/2 program [41]. Nonhydrogen atoms were refined with anisotropic displacement parameters. The hydrogen atoms were fixed at calculated distances and allowed to ride on the parent atoms with U_iso_(H) equal to 1.2U_eq_(C) or 1.5U_eq_(C). Details concerning the determination of crystal structure are gathered in Table 2. CCDC 2298150 contains the Appendix A for this paper.

### 3.4. Synthesis of 9-Methyl-1,10-phenanthroline-2-carboxylic Acid (***2a***). 1.0 g Scale

Molecule **1a** (1.0 g, 4.8 mmol) was added to a solution of NaClO_2_ (5.0 g, 55.6 mmol) in 50 mL of water at room temperature. Then, the temperature was increased to 90 °C. After carrying out the reaction for 6 h at this temperature, the resulting solution was acidified very slowly to pH 2.8 by adding HCl solution (10%). The water was then evaporated under reduced pressure, and the crude product was extracted with MeOH and purified by Soxhlet extraction (MeOH). After the concentration of MeOH, the product was washed with CH_2_Cl_2_ and finally dried over P_4_O_10_.

*9-Methyl-1,10-phenanthroline-2-carboxylic acid* (**2a**) 0.66 g (2.8 mmol, 58%); m.p. > 300 °C; ^1^H NMR (D_2_O/KOD; 400.2 MHz) δ = 2.46 (s, 3H, CH_3_), 6.99 (bs, 2H, aromatic), 7.02 (d, ^3^*J*_H,H_ = 8.2 Hz, 1H, aromatic), 7.50 (d, ^3^*J*_H,H_ = 8.2 Hz, 1H, aromatic), 7.68–7.76 (m, 2H, aromatic), 8.41 (bs, 1H, OH); ^13^C{^1^H} NMR (D_2_O; 100.5 MHz) δ = 23.3, 122.2, 123.7, 124.6, 125.9, 126.8, 128.5, 136.8, 137.0, 142.0, 142.6, 151.2, 158.2, 172.4; HRMS (ESI TOF): *m*/*z* Calcd for C_14_H_9_N_2_O_2_ (M − H)^−^ = 237.0664, Found 237.0662; UV–vis (methanol; λ [nm] (logε)): 314 (3.06), 275 (3.95), 230 (4.10); IR (KBr): $\tilde{\nu }$ = 1628, 1393, 1007, 972, 835 cm^−1^.

10.0 g scale

Molecule **1a** (10.0 g, 48.1 mmol) was added to a solution of NaClO_2_ (50.0 g, 556.2 mmol) in 50 mL of water at room temperature. Then, the temperature was increased to 90 °C. After carrying out the reaction for 6 h at this temperature, the resulting solution was acidified very slowly to pH 2.8 by adding HCl solution (10%). The water was then evaporated under reduced pressure, and the crude product was extracted with MeOH and purified by Soxhlet extraction (MeOH). After the concentration of MeOH, the product was washed with CH_2_Cl_2_ and finally dried over P_4_O_10_.

*4-Chloro-9-methyl-1,10-phenanthroline-2-carboxylic acid* (**2b**) 2.4 g (8.7 mmol, 18%); DSC = 67.4 °C; ^1^H NMR (DMSO-d_6_/D_2_O/KOD; 500.2 MHz) δ = 2.46 (s, 3H, CH_3_), 7.56 (d, ^3^*J*_H,H_ = 8.2 Hz, 1H, aromatic), 7.67 (d, ^3^*J*_H,H_ = 8.4 Hz, 1H, aromatic), 7.97 (s, 1H, aromatic), 8.21 (d, ^3^*J*_H,H_ = 8.3 Hz, 1H, aromatic), 8.43 (d, ^3^*J*_H,H_ = 8.4 Hz, 1H, aromatic); ^13^C{^1^H} NMR (DMSO-d_6_/D_2_O/KOD; 125.8 MHz) δ = 24.3, 124.6, 124.7, 122.7, 125.3, 126.5, 126.6, 127.9, 128.0, 133.3, 136.2, 136.3, 143.9, 145.4, 159.5, 159.9; HRMS (ESI TOF): *m*/*z* Calcd for C_14_H_5_D_3_N_2_O_2_Cl (M − H)^−^ = 274.0463, Found 274.0466.

Oxidation of selected dicarbaldehydes

To a solution of the appropriate hydroxy dicarbaldehyde **1b** or **3**, respectively (1.0 mmol), in water (25 mL) (formic acid for molecule **3**), a solution of NaClO_2_ (5.00 g, 55.6 mmol) in water (25 mL) was slowly added. The reaction was stirred at room temperature for three hours and then at 90 °C for another three hours. The reaction mixture was then concentrated, and the resulting suspension was cooled. The resulting precipitate was filtered off, washed with cold water (3 × 50 mL), and air dried. The product thus obtained was further purified by crystallization from ethanol to obtain the following:*Sodium (E)-3,3-dichloro-2-(2-(hydroxy(methoxy)methylene)-2,3-dihydropyridin-3-yl)acrylate* 0.04 g (0.15 mmol, 15.0%); ^1^H NMR (400.2 MHz, DMSO-d_6_) δ = 3.59 (s, 3H), 7.76 (dd, ^3^*J*_H,H_ = 4.8, 7.8 Hz, 1H), 8.43 (dd, ^3^*J*_H,H_ = 1.7, ^4^*J*_H,H_ = 7.9 Hz, 1H), 9.02 (dd, ^3^*J*_H,H_ = 1.7, ^4^*J*_H,H_ = 4.8 Hz, 1H). ^13^C{^1^H} NMR (DMSO–d_6_; 100.6 MHz) δ = 54.02, 86.56, 90.31, 125.80, 126.89, 134.74, 158.31, 165.55, 168.75, 186.75; HRMS (ESI TOF): *m*/*z* Calcd for C_10_H_8_Cl_2_NNaO_4_ (M)^+^ = 298.9728, Found 299.9632.

*10-Hydroxybenzo[h]quinoline-7,9-dicarboxylic acid* (**3a**) yellowish, 0.09 g (0.32 mmol, 32.4%); m.p._dec_. = 235–245 °C; ^1^H NMR (DMSO–d_6_; 400.2 MHz; 60 °C) δ = 8.13 (t, ^3^*J*_H,H_ = 7.0 Hz, 1H, aromatic), 8.24 (d, ^3^*J*_H,H_ = 9.5 Hz, 1H, aromatic), 8.94 (s, 1H, aromatic), 9.07 (d, ^3^*J*_H,H_ = 8,0 Hz, 1H, aromatic), 9.24 (d, ^3^*J*_H,H_ = 5,4 Hz, 1H, aromatic), 9.36 (d, ^3^*J*_H,H_ = 9,4 Hz, 1H, aromatic), 16.21 (s, 1H, OH); ^13^C{^1^H} NMR (DMSO–d_6_; 100.6 MHz; 60 °C) δ = 112.2, 113.2, 113.4 122.2, 126.1, 127.6, 128.1, 137.5, 138.3, 141.3, 142.1, 143.1 167.5, 167.8, 172.6; HRMS (ESI TOF): *m*/*z* Calcd for C_15_H_9_NO_5_Na (M + Na)^+^ = 306.0378, Found 306.0388; UV–vis (methanol; λ [nm] (logε)): 322 (3,96), 308 (4,03), 255 (4,49), 232 (4,84), 220 (4,85); IR (KBr): $\tilde{\nu }$ = 2928, 1718, 1678, 1514, 1464, 1245, 846 cm^−1^.

Synthesis of 8-hydroxy-2-methylquinoline-5,7-dicarboxylic acid

A solution of 8-hydroxy-2-methylquinoline-5,7-dicarbaldehyde (**1b**) (0.6 g, 2.8 mmol) in nitric acid (65%, 20 mL) was heated to 70 °C for three hours. The mixture was poured onto ice (ca. 40 mL), and a yellow solid precipitated. It was collected by filtration, washed with water, and dried over a vacuum to yield a yellow solid (3.80 g, 86% yield). The crude material obtained was crystallized from methanol to give the following:
*2-Methyl-5,7-dinitroquinolin-8-ol* as a pale yellow powder 0.45 g (1.8 mmol, 65.0%); ^1^H NMR (DMSO-d_6_; 500.2 MHz) δ = 2.94 (s, 3H, CH_3_), 8.14 (d, ^3^*J*_H,H_ = 9.0 Hz, 1H, aromatic), 9.21 (s, 1H, aromatic), 9.66 (d, ^3^*J*_H,H_ = 9.0 Hz, 1H, aromatic); ^13^C{^1^H} NMR (DMSO-d_6_; 125.8 MHz) δ = 19.9, 122.6, 125.0, 129.0, 129.3, 130.8, 137.4, 141.2, 154.9, 161.7; HRMS (ESI TOF): *m*/*z* Calcd for C_10_H_6_N_3_O_5_ (M − H)^−^ = 248.0307, Found 248.0309. IR (KBr): 3510; 3170; 3005; 1594; 1528; 1321; 1260.

Tollens reaction

To the solution composed of AgNO_3_ (16.9 g, 100.0 mmol) in water (100 mL), NaOH (4.0 g, 100.0 mmol) was added, and brown silver(I) oxide was precipitate. Next, aqueous ammonia (25%) was added until the precipitate was dissolved, forming diamminesilver(I), followed by the addition of 8-hydroxy-2-methylquinoline-5,7-dicarbaldehyde (1.1 g, 5.1 mmol). The reaction was stirred at room temperature for 16 h. The resulting precipitate was filtered off, washed with water (3 × 50 mL), and acidified, and the resulting precipitate was filtered off. The obtained yellow solution was evaporated. The product thus obtained was further purified by crystallization from ethanol.

The synthesis of methyl 4,7-dichloro-2,9-dimethyl-1,10-phenanthroline-5-carboxylate was based on our procedure described in the literature [32].

*Methyl 4,7-dichloro-2,9-dimethyl-1,10-phenanthroline-5-carboxylate* (**1d**) beige; DSC m.p. = 175.8 °C; ^1^H-NMR (CDCl_3_; 500.2 MHz) δ = 2.89 (s, 3H, CH_3_), 2.89 (s, 3H, CH_3_), 3.99 (s, 3H, CH_3_), 7.61 (s, 1H, aromatic), 7.62 (s, 1H, aromatic), 8.29 (s, 1H, aromatic); ^13^C{^1^H}-NMR (CDCl_3_; 125.8 MHz) δ = 25.6, 26.0, 53.2, 121.9, 123.5, 124.7, 125.7, 128.6, 141.3, 143.2, 146.6, 146.9, 160.4, 161.5, 169.3; HRMS (ESI TOF): *m*/*z* Calcd for C_16_H_13_N_2_O_2_Cl_2_ (M + H)^+^ = 335.0354, Found 335.0353.*Synthesis of dithiocarboxylic acid* (**5a**): A solution of **1a** (2.9 g, 13.9 mmol) in THF (50 mL) was cooled (−78 °C), and *t*BuLi (11.8 mL; 1.7 M solution in pentane) was added dropwise under argon. The resulting violet solution was stirred for 1 h at ca. −78 °C, and then CS_2_ (5 mL, 6.3 g, 83.1 mmol) was added and stirred for 1 h at ca. −78 °C, and next was stirred for 16 h at r.t. The volatiles were removed in vacuo (16 mmHg). The residue was dissolved in water, acidified by a water solution of hydrochloric acid (10%), and filtered off. The crude product was dissolved in NaOH (10%) water solution and filtered off. The solution obtained was again acidified by a water solution of hydrochloric acid (10%), and the precipitate product was obtained, filtered off, and dried over P_4_O_10_ to yield the following:*2,9-Dimethyl-1,10-phenanthroline-4-carbodithioic acid* (**5a**) (red) 0.3 g (1.0 mmol, 7%), DSC = 204.5 °C; 1H NMR (DMSO-d_6_; 500.2 MHz) δ = 2.97 (s, 3H, CH_2_), 3.01 (s, 3H, CH_2_), 7.99 (d, ^3^*J*_H,H_ = 8.4 Hz, 1H, aromatic), 8.12 (d, ^3^*J*_H,H_ = 8.9 Hz, 1H, aromatic), 8.16 (d, ^3^*J*_H,H_ = 8.9 Hz, 1H, aromatic), 8.35 (s, 1H, aromatic), 8.84 (d, ^3^*J*_H,H_ = 8.3 Hz, 1H, aromatic), 10.47 (s, SH, 1H); ^13^C{^1^H} NMR (DMSO-d_6_; 125.8 MHz) δ = 20.3, 22.7, 125.6, 126.1, 126.5, 126.9, 128.6, 132.2, 133.6, 136.9, 141.4, 153.0, 155.0, 158.5, 247.8; HRMS (ESI TOF): *m*/*z* Calcd for C_15_H_12_N_2_S_2_ (M + H)^+^ = 285.0520, Found 285.0515;

Synthesis of vinyl derivatives 6.

Molecule **1a** or **1h** (5.0 mmol) was dissolved in Ac_2_O (100 mL), followed by the addition of 2-hydroxybenzaldehyde (2.44 g, 20.0 mmol) or 2,4-dihydroxybenzaldehyde (2.76 g, 20.0 mmol), respectively. The reaction was heated under reflux for 24 h. Then, the volatiles were removed in vacuo (16 mmHg). A water solution of hydrochloric acid (10%) was added, and the reaction mixture was stirred under further reflux for 16 h. The crude product was purified by crystallization from ethanol (or methanol) and dried over P_4_O_10_ to yield the following precipitates:*2,9-Bis((E)-2-hydroxystyryl)-5-methyl-1,10-dihydro-1,10-phenanthroline-4,7-dione* (**6a**) yellow ^1^H-NMR (MeOD/KOD; 500.2 MHz) δ = 2.59 (s, 3H, CH_3_), 6.35 (d, ^3^*J*_H,H_ = 10.0 Hz, 1H, aromatic), 6.49 (dd, ^3^*J*_H,H_ = 5.0 Hz, 1H, aromatic), 6.70 (dd, ^3^*J*_H,H_ = 8.3 Hz, ^4^*J*_H,H_ = 1.2 Hz, 1H, aromatic), 6.84 (s, 1H, aromatic), 7.66 (d, ^4^*J*_H,H_ = 1.2 Hz, 1H, aromatic), 7.69 (dd, ^3^*J*_H,H_ = 5.5 Hz, ^4^*J*_H,H_ = 1.2 Hz, 1H, aromatic), 8.24 (d, ^3^*J*_H,H_ = 16.4 Hz, 1H, vinyl), 8.33 (d, ^3^*J*_H,H_ = 16.1 Hz, 1H, vinyl), 8.52 (s, 1H, OH), 10.22 (s, 1H, NH); ^13^C{^1^H}-NMR (MeOD/KOD; 125.8 MHz) δ = 24.3, 109.5, 111.8, 113.6, 115.1, 120.1, 120.3, 121.6, 122.0, 123.3, 124.0, 125.2, 126.7, 126.9, 131.1, 131.6, 131.7, 132.1, 133.2, 139.0, 152.1, 157.6, 161.3, 167.9, 175.0, 175.6, 175.8, 182.0, 182.1; HRMS (ESI TOF): *m*/*z* Calcd for C_29_H_23_N_2_O_4_ (M + H)^+^ = 463.1658, Found 463.1656.*2,9-Bis((E)-2,4-dihydroxystyryl)-5-methyl-1,10-dihydro-1,10-phenanthroline-4,7-dione* (**6b**) fulvous; DSC m.p. = 65.55 °C; HRMS (ESI TOF): *m*/*z* Calcd for C_29_H_23_N_2_O_6_ (M + H)^+^ = 495.1556, Found 495.1555.*((1E,1′E)-(1,10-Phenanthroline-2,9-diyl)bis(ethene-2,1-diyl))bis(2,1-phenylene) bis(hydrogen carbonate)* (**6c**) beige; DSC m.p. = 213.97 °C; 2.4 g (4.8 mmol, 96%), 1H-NMR (DMSO-d_6_; 500.2 MHz) δ = 2.45 (s, 6H, CH_3_), 7.25 (dd, ^3^*J*_H,H_ = 8.0 Hz, ^4^*J*_H,H_ = 1.4 Hz, 2H, aromatic), 7.39 (td, ^3^*J*_H,H_ = 7.0 Hz, ^4^*J*_H,H_ = 1.4 Hz, 2H, aromatic), 7.44 (td, ^3^*J*_H,H_ = 7.6 Hz, ^4^*J*_H,H_ = 1.6 Hz, 2H, aromatic), 7.66 (d, ^3^*J*_H,H_ = 16.3 Hz, 2H, vinyl), 7.96 (d, ^3^*J*_H,H_ = 17.3 Hz, 2H, vinyl), 8.02 (d, ^3^*J*_H,H_ = 8.7 Hz, 3H, aromatic), 8.12 (d, ^3^*J*_H,H_ = 8.4 Hz, 2H, aromatic), 8.48 (d, ^3^*J*_H,H_ = 8.3 Hz, 2H, aromatic), 10.09 (s, 1H, OH); ^13^C{^1^H}-NMR (DMSO-d_6_; 125.8 MHz) δ = 20.8, 121.7, 123.3, 126.2, 126.4, 126.57, 126.62, 128.0, 128.9, 129.6, 131.1, 136.9, 145.2, 148.6, 154.7, 169.3; HRMS (ESI TOF): *m*/*z* Calcd for C_32_H_25_N_2_O_4_ (M + H)^+^ = 501.1814, Found 501.1806.

Molecule **6c** (5.0 mmol) was added to a water solution of hydrochloric acid (10%) and stirred under further reflux for 16 h. Next, a water solution of sodium hydroxide (10%) was added, and the red precipitate was filtered off and purified by crystallization from methanol and dried over P_4_O_10_ to yield the following precipitates:*2,2′-((1E,1′E)-(1,10-Phenanthroline-2,9-diyl)bis(ethene-2,1-diyl))diphenol* (**6d**) red 1.9 g (4.6 mmol, 91%), DSC m.p. = 183.50 °C; ^1^H-NMR (DMSO-d_6_; 500.2 MHz) δ = 6.95 (td, ^3^*J*_H,H_ = 7.4 Hz, ^4^*J*_H,H_ = 1.2 Hz, 2H, aromatic), 7.09 (dd, ^3^*J*_H,H_ = 8.2 Hz, ^4^*J*_H,H_ = 1.2 Hz, 2H, aromatic), 7.28 (ddd, ^3^*J*_H,H_ = 8.5 Hz, ^3^*J*_H,H_ = 7.1 Hz, ^4^*J*_H,H_ = 1.7 Hz, 2H, aromatic), 7.70 (dd, ^3^*J*_H,H_ = 7.9 Hz, ^4^*J*_H,H_ = 1.7 Hz, 2H, aromatic), 8.17 (d, ^3^*J*_H,H_ = 16.4 Hz, 2H, vinyl), 8.20 (s, 2H, aromatic), 8.39 (d, ^3^*J*_H,H_ = 16.5 Hz, 2H, vinyl), 8.56 (d, ^3^*J*_H,H_ = 8.8 Hz, 2H, aromatic), 8.88 (d, 3JH,H = 8.7 Hz, 2H, aromatic); ^1^H-NMR (DMSO-d_6_/KOD; 500.2 MHz) δ = 6.22–6.34 (m, 2H, aromatic), 6.45 (d, ^3^*J*_H,H_ = 17.2 Hz, 1H, vinyl), 6.46 (d, ^3^*J*_H,H_ = 16.8 Hz, 1H, vinyl), 6.85–6.94 (m, 2H, aromatic), 7.37–7.40 (m, 1H, aromatic), 7.41–7.44 (m, 1H, aromatic), 7.49 (dd, ^3^*J*_H,H_ = 5.7 Hz, ^4^*J*_H,H_ = 2.1 Hz, 2H, aromatic), 7.74 (s, 2H, aromatic), 8.09 (d, ^3^*J*_H,H_ = 8.5 Hz, 2H, aromatic), 8.19 (d, ^3^*J*_H,H_ = 16.7 Hz, 2H, vinyl), 8.26 (d, ^3^*J*_H,H_ = 8.5 Hz, 2H, aromatic); ^13^C{^1^H}-NMR (DMSO-d_6_/KOD; 125.8 MHz) δ = 112.3, 120.5, 122.1, 123.4, 125.3, 126.0, 127.2, 127.7, 131.2, 135.1, 137.3, 145.6, 159.0, 170.2; ^13^C{^1^H}-NMR (DMSO-d_6_; 125.8 MHz) δ = 116.6, 119.6, 122.3, 122.4, 123.6, 126.4, 128.1, 129.1, 131.4, 136.8, 136.9, 140.9, 155.2, 157.1, 169.7; HRMS (ESI TOF): *m*/*z* Calcd for C_28_H_21_N_2_O_2_ (M + H)^+^ = 417.1603, Found 417.1598.

### 3.5. Computational Method

Density functional theory calculations were carried out with Gaussian16 software [42] using the B3LYP functional [43,44] and cc-pVTZ basis set for all atoms. Molecular orbitals were visualized using Gaussview 6.0 [45]. C-13 NMR chemical shifts were calculated with Gaussian16 using the gauge-independent atomic orbital (GIAO) method and were plotted relative to the calculated TMS reference (B3LYP/6-311 + G(2d,p)) in Gaussview.

## 4. Conclusions

The presented research has focused on synthesizing water-soluble ligands based on the 1,10-phenanthroline core. Hydrophilic properties were obtained by introducing phenol, carboxylic acid, and dithiocarboxylic acid groups. Using simple and efficient reactions, we have developed a synthetic strategy for the dithiocarboxylic acid **5a** and vinyl (other name styryl) analog **6d** previously unknown representatives of water-soluble ligands based on a 1,10-phenanthroline core. In C-13 NMR spectroscopy, we observed a chemical shift of 248 ppm for the dithiocarboxylic acid group, similar to our previous paper [34,35]. The proton transfer reactions between the pyridine and carboxylic acid, dithiocarboxylic acid, or phenol functional groups or the generally simple protonation or deprotonation of pyridine rings in the 1,10-phenanthroline constitution could be used as a pH indicator. The compounds were further characterized by analytical methods and rationalized on the basis of DFT calculations B3LYP functional.

## Data Availability

The data set presented in this study is available in this article. Appendix A: The following are available online: CCDC 2298150 for **1d** contains the Appendix A for this paper, accessed on 28 April 2023. These data can be obtained free of charge from http://www.ccdc.cam.ac.uk/conts/retrieving.html (accessed on 13 June 2022) or from the Cambridge Crystallographic Data Centre, 12 Union Road, Cambridge CB2 1EZ, UK; Fax: (+44) 1223-336-033; or e-mail: deposit@ccdc.cam.ac.uk.

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
