# Peer review of "Synthesis and Spectroscopic Characterization of Selected Water-Soluble Ligands Based on 1,10-Phenanthroline Core"

_molecules, 2024, doi:10.3390/molecules29061341_

Round 1

Reviewer 1 Report

Comments and Suggestions for Authors

The paper by Nycz et al. reports the synthesis a series of 1,10-phenanthroline derivatives by using synthetic methods alternative to those already present in the literature.

The chemistry of phenanthroline-based compounds is appealing due to their use in variety of fields, from sensing systems to the achievement of biologically active molecules.  At the same time, the procedures reported in this paper appear of interest, due to the milder conditions used with respect to already known synthetic methods.   The compounds appear well characterized. I am positively impressed by this work, and I suggest its publication.

My only comment  concerns the different products obtained for the oxidation of compound 1a, depending on the quantity of reactant used. The author should better clarify the reasons for the different reactivity observed for 1a in the presence of NaClO2.    

Comments on the Quality of English Language

The English language appears. Minor editor revisions could be necessary

Author Response

Dear Editor,

We want to thank the referees whose comments helped us improve our manuscript. Further, we provide detailed comments on the referees’ notes. All changes made in the text are highlighted in yellow color.

Editor's Comments:

Please provide a picture as the graphical abstract when you submit the revised version of your manuscript.

Answer: The suggested graphical abstract has been designed and submitted.

A point-by-point response to comments from Reviewer 1

The paper by Nycz et al. reports the synthesis a series of 1,10-phenanthroline derivatives by using synthetic methods alternative to those already present in the literature.

The chemistry of phenanthroline-based compounds is appealing due to their use in variety of fields, from sensing systems to the achievement of biologically active molecules. At the same time, the procedures reported in this paper appear of interest, due to the milder conditions used with respect to already known synthetic methods. The compounds appear well characterized. I am positively impressed by this work, and I suggest its publication.

My only comment concerns the different products obtained for the oxidation of compound 1a, depending on the quantity of reactant used. The author should better clarify the reasons for the different reactivity observed for 1a in the presence of NaClO2.

Answer: Thanks for this valuable recommendation.

One of the examples of using sodium chlorite (NaClO2) under mild acidic conditions is the Pinnick oxidation of aldehydes, which converts them into corresponding carboxylic acids. The Pinnick oxidation has proven to be both tolerant of sensitive functionalities and capable of reacting with sterically hindered groups. The reaction is highly suited for substrates with many group functionalities. During this reaction, hypochlorous acid (HOCl) is released. The HOCl byproduct, a reactive oxidizing agent, can also cause other undesired side reactions with the organic materials. For example, HOCl can react with double bonds in the organic reactant or product via a halohydrin formation reaction. The presence of molecule 2b with a chlorine atom at the C4 position suggests a mechanism of SNAr, for which there are no examples in the literature. Further systematic work is necessary.

Some improvements in the writing have been made. I have carefully revised the whole manuscript and tried to avoid grammar or syntax errors. Besides, I have asked several skilled authors of English language papers to check the English. Thank you so much for your help. I appreciate it.

Yours sincerely, Jacek Nycz (on behalf of all co-authors)

Reviewer 2 Report

Comments and Suggestions for Authors

Nycz et al. synthesized soluble ligands based on the 1,10-phenanthroline core and analyzed their properties. 1,10-phenanthroline is a material capable of metal coordination or metal chelating and is used in a variety of applications. 1,10-phenanthroline, with its water-soluble properties, can potentially contribute to research and development in application fields. The experimental results in this manuscript are robust and well organized. However, additional information about their potential applications and comparisons to existing soluble 1,10-phenanthroline derivatives would strengthen the manuscript. I support the paper being published after making the few changes below.

1. The authors mentioned newly developed water-soluble 1,10-phenanthroline derivatives as well as previously reported ones. To improve reader readability, I suggest adding a table of properties and sources for the author's newly synthesized water-soluble 1,10-phenanthroline and 1,10-phenanthroline derivatives reported elsewhere. This will be useful for readers to understand readability and research trends in related fields.

2. The size of the figure related to the spectrum in the middle of Figure 2 must be the same.

3. At the beginning of the introduction, “Water-soluble ligands based on 1,10-phenanthroline core are relatively poorly studied 14 compounds. Developing efficient and convenient syntheses of them would result in new interesting applications because of the importance of 1,10-phenanthrolines.” I think this does not sufficiently emphasize the importance of this study. And it seems that the order of the first and second sentences has changed. I propose to describe the research background and research purpose more clearly at the beginning of the Introduction.

4. Line 336-338: “These data can be obtained free of charge via http://www.ccdc.cam.ac.uk/conts/retrieving.html (or from the CCDC, 12 Union Road, Cambridge CB2 1EZ, UK; Fax: +44 1223 336033; E-mail: deposit@ccdc.cam.ac.uk)” is duplicated with lines 515-518. These sentences should go to “Data Availability Statement” on page 16. 

5. Reference DOIs that are included and those that are not are mixed. This must be unified according to the journal format. Also, authors need to carefully check the reference. For example, reference 37 states “ Gaussian 16, Revision C.01 “is included.

Author Response

Dear Editor,

We want to thank the referees whose comments helped us improve our manuscript. Further, we provide detailed comments on the referees’ notes. All changes made in the text are highlighted in yellow color.

Editor's Comments:

Please provide a picture as the graphical abstract when you submit the revised version of your manuscript.

Answer: The suggested graphical abstract has been designed and submitted.

A point-by-point response to comments from Reviewer 2

Nycz et al. synthesized soluble ligands based on the 1,10-phenanthroline core and analyzed their properties. 1,10-phenanthroline is a material capable of metal coordination or metal chelating and is used in a variety of applications. 1,10-phenanthroline, with its water-soluble properties, can potentially contribute to research and development in application fields. The experimental results in this manuscript are robust and well organized.

Answer Thanks for this valuable recommendation.

However, additional information about their potential applications and comparisons to existing soluble 1,10-phenanthroline derivatives would strengthen the manuscript. I support the paper being published after making the few changes below.

  1. The authors mentioned newly developed water-soluble 1,10-phenanthroline derivatives as well as previously reported ones. To improve reader readability, I suggest adding a table of properties and sources for the author's newly synthesized water-soluble 1,10-phenanthroline and 1,10-phenanthroline derivatives reported elsewhere. This will be useful for readers to understand readability and research trends in related fields.

Answer: We have added the suggested table replacing the Scheme 1.

  1. The size of the figure related to the spectrum in the middle of Figure 2 must be the same.

Answer: The suggested correction has been made.

  1. At the beginning of the introduction, “Water-soluble ligands based on 1,10-phenanthroline core are relatively poorly studied 14 compounds. Developing efficient and convenient syntheses of them would result in new interesting applications because of the importance of 1,10-phenanthrolines.” I think this does not sufficiently emphasize the importance of this study. And it seems that the order of the first and second sentences has changed. I propose to describe the research background and research purpose more clearly at the beginning of the Introduction.

Answer: We have added the fragment: “The chemical and physical properties of water-soluble 1,10-phenanthroline derivatives with sulfonate, carboxylic, phosphonic acid, or hydroxylic groups in their constitution, have been widely studied and used for separation of metallic atoms and capillary electrophoresis, as well as for the development of bio-inorganic probes [5]. Furthermore, these hydrophilic groups' presence allows researchers to conduct in vitro cytotoxicity tests and assess the compounds' biological activity [6]. In addition, complexes with 1,10-phenanthroline derivatives functionalized with sulfonate, carboxylic, phosphonic acid, or hydroxylic groups are capable of binding to metal oxide surfaces [7].”

Additionally, we add the ref. 5, 7, 22, 24, 25 and 29.

  1. Line 336-338: “These data can be obtained free of charge via http://www.ccdc.cam.ac.uk/conts/retrieving.html (or from the CCDC, 12 Union Road, Cambridge CB2 1EZ, UK; Fax: +44 1223 336033; E-mail: deposit@ccdc.cam.ac.uk)” is duplicated with lines 515-518. These sentences should go to “Data Availability Statement” on page 16. 

Answer: The suggested correction has been made.

  1. Reference DOIs that are included and those that are not are mixed. This must be unified according to the journal format. Also, authors need to carefully check the reference. For example, reference 37 states “ Gaussian 16, Revision C.01 “is included.

Answer: The suggested correction has been made.

Some improvements in the writing have been made. I have carefully revised the whole manuscript and tried to avoid grammar or syntax errors. Besides, I have asked several skilled authors of English language papers to check the English. Thank you so much for your help. I appreciate it.

Yours sincerely, Jacek Nycz (on behalf of all co-authors)

Round 2

Reviewer 2 Report

Comments and Suggestions for Authors

The authors have addressed the reviewer's concerns, and the revised manuscript is now suitable for publication.